

**Temporal dynamics of surface ocean carbonate chemistry in response**
**to natural and simulated upwelling events during the 2017 coastal El**
**Niño near Callao, Peru**
Shao-Min Chen[1, 2], Ulf Riebesell[1], Kai G. Schulz[3], Elisabeth von der Esch[1], Eric P. Achterberg[1], and
Lennart T. Bach[4]
[1]GEOMAR Helmholtz Centre for Ocean Research Kiel, Kiel, Germany
[2]Department of Earth and Environmental Sciences, Dalhousie University, Halifax, Canada
[3]Centre for Coastal Biogeochemistry, School of Environment, Science and Engineering, Southern Cross University, Lismore,
Australia
[4]Institute for Marine and Antarctic Studies, University of Tasmania, Tasmania, Australia
*Correspondence to*: Shao-Min Chen (shaomin.chen@dal.ca)



**Abstract.** Oxygen minimum zones (OMZs) are characterized by enhanced carbon dioxide ($CO_2$) levels and low pH and are
being further acidified by uptake of anthropogenic atmospheric $CO_2$. With ongoing intensification and expansion of OMZs
due to global warming, carbonate chemistry conditions may become more variable and extreme, particularly in the Eastern
Boundary Upwelling Systems. In austral summer (Feb-Apr) 2017, a large-scale mesocosm experiment was conducted in the
coastal upwelling area off Callao (Peru) to investigate the impacts of on-going ocean deoxygenation on biogeochemical
processes, coinciding with a rare coastal El Niño event. Here we report on the temporal dynamics of carbonate chemistry in
the mesocosms and surrounding Pacific waters over a continuous period of 50 days with high temporal resolution observations
(every 2nd day). The mesocosm experiment simulated an upwelling event in the mesocosms by addition of nitrogen (N)-
deficient and $CO_2$-enriched OMZ water. Surface water in the mesocosms was acidified by the OMZ water addition, with $pH_T$
lowered by 0.1-0.2 and $pCO_2$ elevated to above 900 µatm. Thereafter, surface $pCO_2$ quickly dropped to near or below the
atmospheric level (405.22 µatm in 2017, NOAA/GML) mainly due to enhanced phytoplankton production with rapid $CO_2$
consumption. Further observations revealed that the dominance of dinoflagellate *Akashiwo sanguinea* and contamination of
bird excrements played important roles in the dynamics of carbonate chemistry in the mesocosms. Compared to the simulated
upwelling, natural upwelling events in the surrounding Pacific waters occurred more frequently with sea-to-air $CO_2$ fluxes of
4.2-14.0 mmol C $m^{-2}$ $d^{-1}$. The positive $CO_2$ fluxes indicated our site was a local $CO_2$ source during our study, which may have
been impacted by the coastal El Niño. However, our observations of DIC drawdown in the mesocosms suggests that $CO_2$
fluxes to the atmosphere can be largely dampened by biological processes. Overall, our study characterized carbonate
chemistry in near-shore Pacific waters that are rarely sampled in such temporal resolution and hence provided unique insights
into the $CO_2$ dynamics during a rare coastal El Niño event.
**1 Introduction**
One of the most extensive oxygen minimum zones (OMZs) in the global ocean can be found off central/northern Peru (4 - 16°
S, Chavez and Messié, 2009). High biological productivity is stimulated by permanent upwelling of cold, nutrient-rich water
to the surface supporting a remarkable fish production off Peru (Chavez et al., 2008; Montecino and Lange, 2009; Albert et
al., 2010). The high primary production also leads to enhanced remineralization of sinking organic matter in subsurface waters
which depletes dissolved oxygen ($O_2$) and creates an intense and shallow OMZ (Chavez et al., 2008). The depletion of $O_2$ in
OMZs plays an important role in the global nitrogen (N) cycle, accounting for 20 - 40% N loss in the ocean (Lam et al., 2009;
Paulmier and Ruiz-Pino, 2009). Denitrification and anammox processes that occur in $O_2$ depleted waters remove N from the
ocean and produce an N deficit and hence phosphorus (P) excess with respect to the Redfield ratio (C:N:P = 106:16:1) in the
water column (Redfield, 1963; Deutsch et al., 2001; Deutsch et al., 2007; Hamersley et al., 2007; Galán et al., 2009; Lam et
al., 2009). Upwelling of this N-deficient water has been found to control the surface-water nutrient stoichiometry and thus
influence phytoplankton growth and community compositions (Franz et al., 2012; Hauss et al., 2012).
Apart from being N-deficient, the OMZ waters are also characterized by enhanced carbon dioxide ($CO_2$) concentrations and
low pH from respiratory processes and are further acidified by uptake of anthropogenic atmospheric $CO_2$ (Feely et al., 2008;
Friederich et al., 2008; Paulmier et al., 2008; Paulmier et al., 2011). Accordingly, surface water carbonate chemistry is
influenced by upwelling of $CO_2$-enriched OMZ water (Van Geen et al., 2000; Capone and Hutchins, 2013). The upwelled
$CO_2$-enriched OMZ water can give rise to surface $CO_2$ levels >1,000 µatm, pH values as low as 7.6, and under-saturation for
the calcium carbonate mineral aragonite (Feely et al., 2008; Hauri et al., 2009). As a result, there is a significant flux of $CO_2$
from the ocean to the atmosphere off Peru, which is further facilitated by surface ocean warming, making the Peruvian
upwelling region a year-round $CO_2$ source to the atmosphere (Friederich et al., 2008). In contrast, rapid utilization of upwelled





$CO_2$ and nutrients by phytoplankton can occasionally deplete surface $CO_2$ below atmospheric equilibrium and dampen the $CO_2$
outgassing (Van Geen et al., 2000; Friederich et al., 2008; Loucaides et al., 2012). The enhanced primary production in turn
contributes to increasing export of organic matter, enhanced bacterial respiration, $O_2$ consumption and $CO_2$ production at
depth. Such a positive feedback may determine the intensity of the underlying OMZ and promote carbon (C) preservation in
marine sediments (Dale et al., 2015).
In response to reduced $O_2$ solubility and enhanced stratification induced by global warming, OMZs have been intensifying and
expanding over the past decades (Stramma et al., 2008; Fuenzalida et al., 2009; Stramma et al., 2010). Based on regional
observations and model projections, a decline in dissolved $O_2$ concentrations has been reported for most regions of the global
ocean (Matear et al., 2000; Matear and Hirst, 2003; Whitney et al., 2007; Stramma et al., 2008; Keeling et al., 2009; Bopp et
al., 2013; Schmidtko et al., 2017; Oschlies et al., 2018). The vertical expansion of OMZs represents shoaling of $CO_2$-enriched
seawater, which has become further enriched by oceanic uptake of anthropogenic $CO_2$ (Doney et al., 2012; Gilly et al., 2013;
Schulz et al., 2019). Since biogeochemical processes in OMZs are directly linked to the C cycle and control surface nutrient
stoichiometry, with on-going ocean warming and acidification, the deoxygenation may have cascading effects on plankton
productivity and composition, C uptake, and food web functioning (Keeling et al., 2009; Gruber, 2011; Doney et al., 2012;
Gilly et al., 2013; Levin and Breitburg, 2015). Therefore, it is important to monitor the changes in $CO_2$ when investigating the
effects of deoxygenation on marine ecosystems.
To investigate the potential impacts of upwelling on pelagic biogeochemistry and natural plankton communities in the Peruvian
OMZ, a large-scale *in situ* mesocosm study was carried out in the coastal upwelling area off Peru. An upwelling event was
simulated in the mesocosms by addition of OMZ waters collected from two different locations where the OMZ was considered
to contain different nutrient concentrations and N:P ratios. The ecological and biogeochemical responses in the mesocosms
were monitored and compared with those influenced by natural upwelling events in the ambient coastal water surrounding the
mesocosms. As part of this collaborative research project, questions specific to the present paper were: (1) How does surface
water carbonate chemistry respond to an upwelling event?; and (2) How does upwelled OMZ water with different chemical
signatures modulate surface water carbonate chemistry? The current study will mainly focus on the temporal changes in surface
water carbonate chemistry within the individual mesocosms, including observations made in the ambient Pacific water and a
local estimate of air-sea $CO_2$ exchange, together with the influence by a rare coastal El Nino event (Garreaud, 2018). This
provides first insights into how inorganic C cycling links to chemical signatures of OMZ waters in a natural plankton
community and its implications for ongoing environmental changes.
**2 Material and methods**
**2.1 Study site**
The experiment was conducted in the framework of the Collaborative Research Center 754 "Climate-Biogeochemistry
Interactions in the Tropical Ocean" (www.sfb754.de/en) and in collaboration with the Instituto del Mar del Peru (IMARPE)
in Callao, Peru (Fig. 1a). The coastal area off Callao lies within the Humboldt Current System and is influenced by wind-
induced coastal upwelling (Bakun and Weeks, 2008).
**2.2 Mesocosm setup**
Eight "Kiel Off-Shore Mesocosms for Future Ocean Simulations" (KOSMOS) units (M1-M8), extending 19 m below the sea
surface, were deployed by the research vessel *Buque Armada Peruana (BAP) Morales* and moored at 12.06° S, 77.23° W in





the coastal upwelling area off Callao, Peru (Fig. 1a) on February 23$^{rd}$, 2017 (late austral summer). The technical design of
these sea-going mesocosms is described by Riebesell et al. (2013). For a more detailed description of the mesocosm
deployment and maintenance in this study, please refer to Bach et al. (2020a).
The mesocosm bags were filled with surrounding seawater through the upper and lower openings. Both openings were covered
by screens with a mesh size of 3 mm to avoid enclosing larger organisms such as fish. The mesocosm bags were left open
below the water surface for two days, allowing free exchange with surrounding coastal water. On February 25$^{th}$, mesocosm
bags were closed with the screens removed, tops pulled above the sea surface and bottoms sealed with 2-m long conical
sediment traps (Fig. 1b). The experiment started with the closure of the mesocosms (day 0) and lasted for 50 days. Each
mesocosm bag enclosed a seawater volume of ~54 m$^3$. After the bags were closed, daily or every-2$^{nd}$-day sampling was
performed to monitor the initial conditions of the enclosed water before simulating an upwelling event on day 11 and 12 (see
Sect. 2.4 for details).

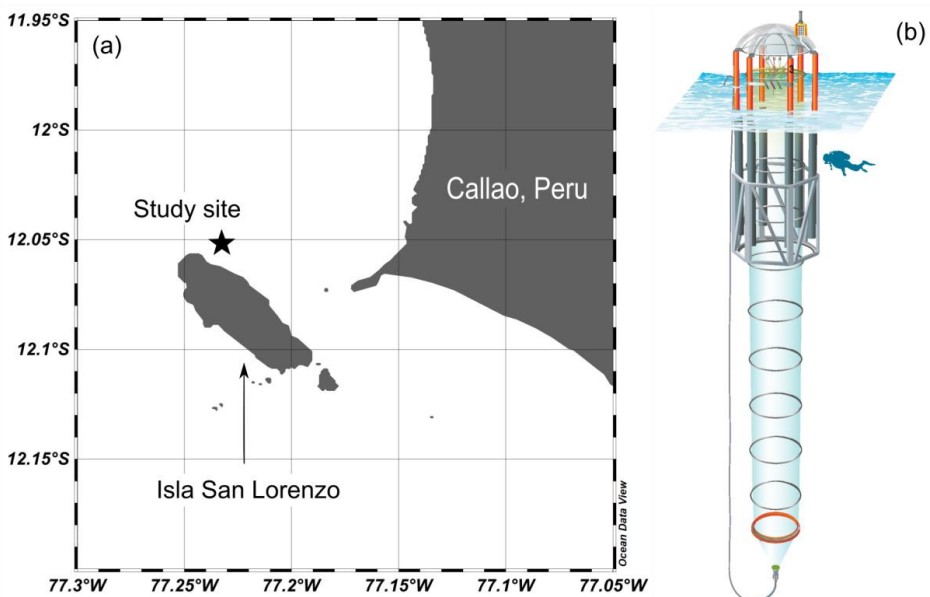

**Figure 1 The study site of the mesocosm experiment (a) created and modified using Ocean Data View (Schlitzer, Reiner, Ocean**
**Data View, odv.awi.de, 2021) and a schematic illustration of a KOSMOS mesocosm unit (b). We acknowledge reprint permission**
**from the AGU as parts of this drawing was used for a publication by Bach et al. (2016). The star symbol marks the approximate**
**location of mesocosm deployment.**
**2.3 Simulated upwelling and salt addition**
To simulate an upwelling event in the mesocosms, OMZ-influenced waters were collected from the nearby coastal area and
added to the mesocosms. Two OMZ water masses were collected at Station 1 (12° 01.70' S, 77° 13.41' W) at a depth of ~30
m and at Station 3 (12° 02.41' S, 77° 22.50' W) at a depth of ~70 m respectively using a deep-water collection system as
described by Taucher et al. (2017). These two water masses were sampled for chemical and biological variables as the
mesocosms (see Sect. 2.4). The OMZ water collected from Station 3 had a dissolved inorganic nitrogen (DIN) concentration
of 4.3 µmol L$^{-1}$ (denoted as "Low DIN" in this paper) and was added to M2, M3, M6 and M7. The OMZ water from Station 1
had a DIN of 0.3 µmol L$^{-1}$ (denoted as "Very low DIN" in this paper) and was added to M1, M4, M5 and M8. Before OMZ
water addition, approximately 9 m$^3$ of seawater were removed from 11-12 m of each mesocosm on March 5$^{th}$ (day 8). During
the night of March 8$^{th}$ (day 11), ~10 m$^3$ of OMZ water were added to 14-17 m of each mesocosm. On March 9$^{th}$ (day 12), ~10
m$^3$ of seawater were removed from 8-9 m followed by an addition of ~12 m$^3$ OMZ water to 0-9 m of each mesocosm.



To maintain a low-$O_2$ bottom layer in the mesocosms and avoid convective mixing induced by heat exchange with the
surrounding Pacific, 69 L of a concentrated sodium chloride (NaCl) brine solution were added to the bottom of each mesocosm
(10-17 m) on day 13, which increased the bottom salinity by ~0.7 units. Since then, turbulent mixing induced by sampling
activities continuously interrupted the artificial halocline. Hence, on day 33, 46 L of the NaCl brine solution were added again
to the bottom of each mesocosm (12.5-17 m), which increased the bottom salinity by ~0.5 units. At the end of the experiment
after the last sampling (day 50), 52 kg of NaCl brine was added again to each mesocosm to calculate the enclosed seawater
volume from a measured salinity change by ~0.2 units (see Czerny et al., 2013 and Schulz et al., 2013 for details). The average
final volume for each mesocosm bag was calculated at ~54 m$^3$. With known sampling volumes and deep-water addition
volumes during the experiment, the enclosed volumes of each mesocosm on each sampling day could be calculated. The NaCl
solution for the halocline establishments had been prepared in Germany by dissolving 300 kg of food industry grade NaCl
(free of anti-caking agents) in 1000 L deionized water (Milli-Q, Millipore) and purified with ion exchange resin (LewawitTM
MonoPlus TP260®, Lanxess, Germany) to minimize potential contaminations with trace metals (Czerny et al., 2013). The
NaCl solution for the volume determination was produced on site using locally purchased table salt. For a more detailed
description of OMZ water and salt additions, please refer to Bach et al. (2020a).
**2.4 Sampling procedures and CTD operations**
Sampling was carried out in the morning (7 a.m.-11 a.m. local time) daily or every 2$^{nd}$ day throughout the entire experimental
period. Depth-integrated samples were taken from the surface (0-10 m for day 3-28) and bottom layer (10-17 m for day 3-28)
of the mesocosms and the surrounding coastal water (named "Pacific") using a 5-L integrating water sampler (IWS, HYDRO-
BIOS, Kiel). Due to the deepening of the oxycline as observed from the CTD profiles, the sampling depth for the surface was
adjusted to 0-12.5 m while that for the bottom was changed to 12.5-17 m from day 29 until the end of the experiment (day 50).
For gas-sensitive variables such as pH and dissolved inorganic carbon (DIC), 1.5 L of seawater from each integrated depth in
each mesocosm were taken directly from the fully-filled 5-L integrating water sampler. Clean polypropylene sampling bottles
(rinsed with deionized water in the laboratory; Milli-Q, Millipore) were pre-rinsed with sample water immediately prior to
sampling. Bottles were filled from bottom to top using pre-rinsed Tygon tubing with overflow of at least one sampling bottle
volume (1.5 L) to minimize the impact of $CO_2$ air-water gas exchange. Nutrient samples were collected into 250 ml
polypropylene bottles using pre-rinsed Tygon tubing (see Bach et al., 2020a for details). Sample containers were stored in cool
boxes for ~3 hours, protected from sunlight and heat before being transported to the shore. Once in the lab, sample water was
sterile-filtered by gentle pressure using syringe filters (0.2 μm pore size), Tygon tubing and a peristaltic pump to remove
particles that may cause changes to seawater carbonate chemistry (Bockmon and Dickson, 2014). For DIC measurements, the
water was filtered from the bottom of the 1.5-L sample bottle into 100-ml glass-stoppered bottles (DURAN) with an overflow
of at least 100 ml to minimize contact with air. Once the glass bottle was filled with sufficient overflow, it was immediately
sealed without headspace using a round glass stopper. This procedure was repeated to collect a second bottle (100 ml) of
filtered water for pH measurements. The leftover seawater was directly filtered into a 500 ml polypropylene bottle for total
alkalinity (TA) measurements (non-gas-sensitive). Filtered DIC and pH samples were stored at 4 ℃ in the dark and TA samples
were at room temperature in the dark until further analysis. Samples were analysed for DIC and pH on the same day of
sampling, while TA was determined overnight (see Sect. 2.5 for analytical procedures).
CTD casts were performed with a multiparameter logging probe (CTD60M, Sea and Sun Technology) in the mesocosms and
Pacific on every sampling day. From the CTD casts, profiles of salinity, temperature, pH, dissolved $O_2$, chlorophyll *a* (chl *a*)
and photosynthetically active radiation were obtained (see Schulz and Riebesell, 2013 and Bach et al., 2020a for details).



**2.5 Carbonate chemistry and nutrient measurements**
Total alkalinity was determined at room temperature (22-32℃) by a two-stage open-cell potentiometric titration using a
Metrohm 862 Compact Titrosampler, Aquatrode Plus (Pt1000) and a 907 Titrando unit in the IMARPE laboratory following
Dickson et al. (2003). The acid titrant was prepared by preparing a 0.05 mol kg$^{-1}$ hydrochloric acid (HCl) solution with an
ionic strength of ca. 0.7 mol kg$^{-1}$ (adjusted by NaCl). Approximately 50-grams of sample water from each sample was weighed
into the titration cell with the exact weight recorded (precision: 0.0001 g). After the two-stage titration, the titration data
between a pH of ~3.5 and 3 was fitted to a modified non-linear Gran approach described in Dickson et al. (2007) using
MATLAB (The MathWorks). The results were calibrated against certified reference materials (CRMs) batch 142 (Dickson,
2010) measured on each measurement day. In this paper, measured TA values refer to the measured values that have been
calibrated against the CRM.
Seawater $pH_T$ (total scale) was determined spectrophotometrically by measuring the absorbance ratios after adding the
indicator dye m-cresol purple (mCP) as described in Carter, et al. (2013). Before measurements, samples were acclimated to
25.0°C in a thermostatted bath. The absorbance of samples with mCP was determined on a Varian-Cary 100 double-beam
spectrophotometer (Varian), scanning between 780 and 380 nm at 1-nm resolution. During the spectrophotometric
measurement, the temperature of the sample was maintained at 25.0°C by a water-bath connected to the thermostatted 10-cm
cuvette. The $pH_T$ values were calculated from the baseline-corrected absorbance ratios and corrected for *in situ* salinity
(obtained from CTD casts) and pH change caused by dye addition (using the absorbance at the isosbestic point, i.e. 479 nm)
as described in Dickson et al. (2007). To minimize potential $CO_2$ air-water gas exchange, a syringe pump (Tecan Cavro XLP)
was used for sample/dye mixing and cuvette injection (see Schulz et al. 2017 for details). For the dye correction, a batch of
sterile filtered seawater of known salinity was prepared. The $pH_T$ was determined once for an addition of 7 ul of dye and once
of 25 ul at five pH levels (raised to 7.95 with NaOH and lowered to 7.74, 7.58, 7.49 and 7.36 with HCl stepwise). The pH
change resulting from the dye correction addition was calculated from the change in measured absorbance ratio for each pair
of dye additions (see Clayton and Byrne, 1993 and Dickson et al., 2007 for details). The dye-corrected $pH_T$ values measured
at 25.0°C and atmospheric pressure were then re-calculated for *in situ* temperature and pressure as determined by CTD casts
(averaged over 0-10/12.5 m for surface and 10/12.5-17 m for bottom). For carbonate chemistry speciation calculations (see
Sect. 2.6), the dye-corrected $pH_T$ values were used as one of the input parameters.
Dissolved inorganic carbon was measured by infrared absorption using a LI-COR LI-7000 on an AIRICA system
(MARIANDA, Kiel, see Taucher et al., 2017 and Gafar and Schulz, 2018 for details). The results were calibrated against
CRMs batch 142 (Dickson, 2010). Unfortunately, due to a malfunctioning of the AIRICA system, we obtained measured DIC
data only up to March 7th (day 10). Therefore, measured TA and $pH_T$ were used for calculations of carbonate system
parameters at *in situ* temperature and salinity but we used DIC measurements from day 3-10 for consistency checks of
calculated carbonate chemistry parameters. In this paper, measured DIC values refer to the measured values that have been
calibrated against the CRM.
Inorganic nutrients were analyzed colorimetrically ($NO_3^-$, $NO_2^-$, $PO_4^{3-}$ and $Si(OH)_4$) and fluorimetrically ($NH_4^+$) using a
continuous flow analyzer (QuAAtro AutoAnalyzer with integrated photometers, SEAL Analytical) connected to a fluorescence
detector (FP-2020, JASCO). All colorimetric methods were conducted according to Murphy and Riley (1962), Mullin and
Riley (1955a, b) and Morris and Riley (1963) and corrected following the refractive index method developed by Coverly et al.
(2012). For details of the quality control procedures, see Bach et al. (2020a).



**2.6 Carbonate chemistry speciation calculations and propagated uncertainties**

Calculations of carbonate chemistry parameters (*in situ* $pH_T$, DIC, $pCO_2$, and calcium carbonate saturation state for calcite and aragonite) were performed with the Excel version of CO2SYS (Version 2.1, Pierrot et al., 2006) using K1 and K2 dissociation constants from Mehrbach et al., (1973) which were refitted by Lueker et al. (2000). The dissociation constant for $KHSO_4$ from Dickson (1990) and for total boron from Uppström (1974) were applied in the calculations (see Orr et al., 2015 for details). The observed $pH_T$ and TA as well as inorganic nutrient concentration (phosphate and silicic acid) were used as input $CO_2$ system parameters. *In situ* salinity and temperature were obtained by CTD casts and averaged over surface (0-10 m or 0-12.5 m) and bottom (10-17 m or 12.5-17 m) waters for each sampling day. In situ pressure was approximated for surface (5 dbar) and bottom (13.5 or 14.75 dbar) waters. For details of calculation procedures and choices of constants, see Lewis et al. (1998) and Orr et al. (2015).

To evaluate the performance of $pH_T$ and TA measurements, quality control procedures were performed. First, standard deviations of $pH_T$ measurements were graphed over time. Measured TA values of a control sample (CRM batch 142, Dickson, 2010) were plotted over time, compared to the warning and control limits calculated from their mean and standard deviation (for details please see Dickson et al., 2007) as well as the certified value of the CRM. To compute a range control chart for the evaluation of measurement repeatability, the absolute difference between duplicate measurements of CRMs on each sampling day was calculated and plotted over time, compared to the warning and control limits calculated from their mean and standard deviation (for details see Dickson et al., 2007).

We used the R package *seacarb* with a Gaussian approach and an input variable pair ($pH_T$, TA) to calculate uncertainties for calculated $CO_2$ system parameters (Orr et al. 2018; Gattuso et al. 2020). The contribution of input uncertainties in nutrient concentrations and *in situ* salinity and temperature to the uncertainties in the CO2SYS-based calculations are often small (< 0.1%; Orr et al. 2018) so they were not considered in our propagation. The input uncertainties of $pH_T$ and TA were estimated based on our measurements (Table 1). Standard uncertainties include random and systematic errors. For TA, systematic errors were removed by calibrating the measured results using CRMs (see Sect. 2.5). Hence, the random error of TA, estimated by the averaged standard deviation of all the CRM measurements (4.4 µmol $kg^{-1}$; n = 62), was used as the standard uncertainty. For $pH_T$, an uncertainty of 0.01 was used as the standard uncertainty. Due to the unavailability of CRMs that correct for systematic error in pH measurements, the standard deviations of repeated measurements (0.0012; n = 377) only accounted for the random components of standard uncertainties (Orr et al. 2018). Therefore, we used 0.01 in our uncertainty propagation as an approximation of the total standard uncertainty for $pH_T$, which has been used in previous assessments (Orr et al. 2018).

**Table 1: Standard uncertainties of $pH_T$ and TA estimated based on our measurements are denoted by u(pHT) and u(TA). Based on u($pH_T$) and u(TA), propagated uncertainties were estimated for each data point in R and averaged for each reported variable (µ), with Standard deviation (σ), minimum (min) and maximum (max) values presented. The relative percentage (%) of propagated standard uncertainties were calculated by dividing the propagated uncertainty by the corresponding data point and averaged for each reported variable (µ), with σ, min and max values presented.**

| $u(pH_T)$ | u(TA) | | $\Delta pCO_2$ | | $\Delta$DIC | | $\Delta\Omega_{Ar}$ | | $\Delta\Omega_{Ca}$ | |
|---|---|---|---|---|---|---|---|---|---|---|
| | µmol $kg^{-1}$ | | µatm | % | µmol $kg^{-1}$ | % | | % | | % |
| 0.01 | 4.4 | µ | 35.94 | 3.8 | 6.63 | 0.3 | 0.08 | 5.1 | 0.13 | 5.1 |
| | | σ | 12.60 | 0.3 | 0.80 | 0.0 | 0.03 | 0.3 | 0.05 | 0.3 |
| | | min | 15.07 | 3.2 | 5.88 | 0.3 | 0.04 | 4.4 | 0.07 | 4.4 |
| | | max | 62.84 | 4.8 | 8.72 | 0.4 | 0.16 | 5.8 | 0.24 | 5.8 |

The air-sea flux of $CO_2$ ($FCO_2$, mmol C $m^{-2}$ $d^{-1}$) in the Pacific was determined based on

$$FCO_2 = k\,K_0\,\Delta pCO_2 \qquad (1)$$



where k is the gas transfer velocity parameterized as a function of wind speed, $K_0$ is the solubility of $CO_2$ in seawater dependent
on *in situ* salinity and temperature (Weiss, 1974), and $\Delta pCO_2$ is the difference between $pCO_2$ in the surface water and in the
atmosphere (Wanninkhof 2014). Wind data were averaged over 2 sampling days for the sampling location from a satellite-
derived gridded dataset (GLDAS Model, near surface wind speed, 0.25 x0.25 degrees, 3 hour temporal resolution, 12.375° to
11.875°S, 77.375° to 76.875°W), obtained from NASA Giovanni (Rodell et al., 2004; Beaudoing and Rodell, 2020). *In situ*
salinity and temperature were obtained from the CTD casts (see Sect. 2.4). Calculated $pCO_2$ based on ($pH_T$, TA) and an
estimated atmospheric $pCO_2$ of 405.22 μatm (referenced to year 2017, NOAA/GML) were used in the air-sea flux estimation.
**3 Results**
**3.1 Responses of surface layer nutrient concentrations**
The OMZ-influenced water masses were collected from two locations and added to the mesocosms to simulate an upwelling
event (see Sect. 2.3). The two water masses were named "Low DIN" and "Very low DIN" respectively based on their DIN
concentrations (Table 2). Both water masses shared similar silicic acid (Si) and phosphate ($PO_4^{3-}$) concentrations but differed
in DIN concentration. The "Low DIN" water had a DIN concentration of 4.3 μmol $L^{-1}$, 14 times as high as that of the "Very
low DIN" water (0.3 μmol $L^{-1}$; Table 2).





**Table 2: Inorganic nutrient concentrations of the two collected deep-water masses. Please note that DIN is the sum of nitrate, nitrite**
**and ammonium. P is phosphate. Si is silicic acid. Color codes denote the two water masses and are applied to the mesocosms treated**
**with respective water masses in the following figures and tables.**

| Water mass | Si ($\mu$mol L$^{-1}$) | DIN ($\mu$mol L$^{-1}$) | PO$_4^{3-}$ ($\mu$mol L$^{-1}$) | N: P ratio (mol: mol) |
|---|---|---|---|---|
| Low DIN | 19.6 | 4.3 | 2.5 | 1.7 |
| Very low DIN | 17.4 | 0.3 | 2.6 | 0.1 |

On day 10 before OMZ water addition, the average surface DIN concentration of the two treatment groups were similar (3.4
$\mu$mol L$^{-1}$), but lower than that in the Pacific (9.8 $\mu$mol L$^{-1}$; Table 3). Surface layer DIN concentration in the mesocosms ranged
between 2.0 and 6.0 $\mu$mol L$^{-1}$ before OMZ water addition (Fig. 2a). The addition of OMZ water elevated surface DIN in the
"Low DIN" mesocosms to 3.6-6.4 $\mu$mol L$^{-1}$ but lowered that in the "Very low DIN" to 0.9-2.0 $\mu$mol L$^{-1}$. The average surface
DIN concentration in the "Very low DIN" decreased to 1.6 $\mu$mol L$^{-1}$ while the "Low DIN" slightly increased to 4.7 $\mu$mol L$^{-1}$
(Table 3), followed by a sharp depletion on day 16 except for M3. M3 received the highest input of DIN (6.4 $\mu$mol L$^{-1}$) and
was not depleted until day 24. Despite several small peaks in M3, M4, M5 and M6 ($\leq$ 1.6 $\mu$mol L$^{-1}$), surface DIN concentration
in the mesocosms were at around limits of detection (LODs: NH$_4^+$ = 0.063 $\mu$mol L$^{-1}$, NO$_2^-$ = 0.054 $\mu$mol L$^{-1}$, NO$_3^-$ = 0.123
$\mu$mol L$^{-1}$) most of the time after depletion. A slight rise could be observed from day 44 towards the last sampling day (day 48).
In the Pacific, surface layer DIN concentration was mostly greater than 5 $\mu$mol L$^{-1}$ (except on day 16 and 18) and became
considerably higher during the second half of the experiment (> 10 $\mu$mol L$^{-1}$ for day 26-44; Fig. 2a).
**Table 3: DIN concentration ($\mu$mol L$^{-1}$) in the surface layer of each mesocosm (M1-M8) and the average DIN concentration ($\mu$mol L$^{-1}$) for each treatment ("Low DIN" and "Very Low DIN", n = 4) before (t10) and after deep water addition (t13). The DIN**
**concentration in the surface Pacific water is also shown. Color codes and symbols denote the respective mesocosm in the following**
**figures.**

| | M1 | M2 | M3 | M4 | M5 | M6 | M7 | M8 | Low DIN | Very Low DIN | Pacific |
|---|---|---|---|---|---|---|---|---|---|---|---|
| t10 | 3.7 | 2.2 | 5.0 | 3.3 | 3.9 | 3.4 | 3.2 | 2.6 | 3.4 ± 1.2 | 3.4 ± 0.5 | 9.8 |
| t13 | 1.8 | 3.6 | 6.4 | 2.0 | 1.6 | 4.7 | 4.0 | 0.9 | 4.7 ± 1.3 | 1.6 ± 0.5 | 9.2 |
| | ■ | ■ | ● | ● | ▲ | ▲ | ▼ | ▼ | | | ● |

Surface layer PO$_4^{3-}$ concentrations in the mesocosms initially ranged between 1.1 and 1.5 $\mu$mol L$^{-1}$ and were elevated by OMZ
water addition to around 1.9 $\mu$mol L$^{-1}$ (Fig. 2b). Thereafter, PO$_4^{3-}$ exhibited a slow but steady decline until the end of the study
with a slightly higher decrease in "Low DIN" mesocosms (blue symbols; Fig. 2b). Throughout the study, PO$_4^{3-}$ in the
mesocosms was never lower than 1.1 $\mu$mol L$^{-1}$. Surface layer PO$_4^{3-}$ in the Pacific was generally higher, fluctuating between
1.4 and 2.9 $\mu$mol L$^{-1}$. In the mesocosms, enhanced chl *a* concentrations were observed at depths shallower than 5 m and below
15 m before OMZ water addition (Fig. 2c). Following OMZ water addition, a chl *a* maximum occurred at ~10 m and persisted
until day 40, except for M3 and M4 with a 1-week delayed increase in the former and a lack of bloom in the latter (Fig. 2c).
After day 40, chl a concentrations in all mesocosms (except for M4) increased to 12-38 $\mu$g L$^{-1}$ with a bloom occurring in 0-10
m (Fig. 2c) Throughout the study, a chl *a* maximum was continuously observed above 10 m in the Pacific (Fig. 2c).


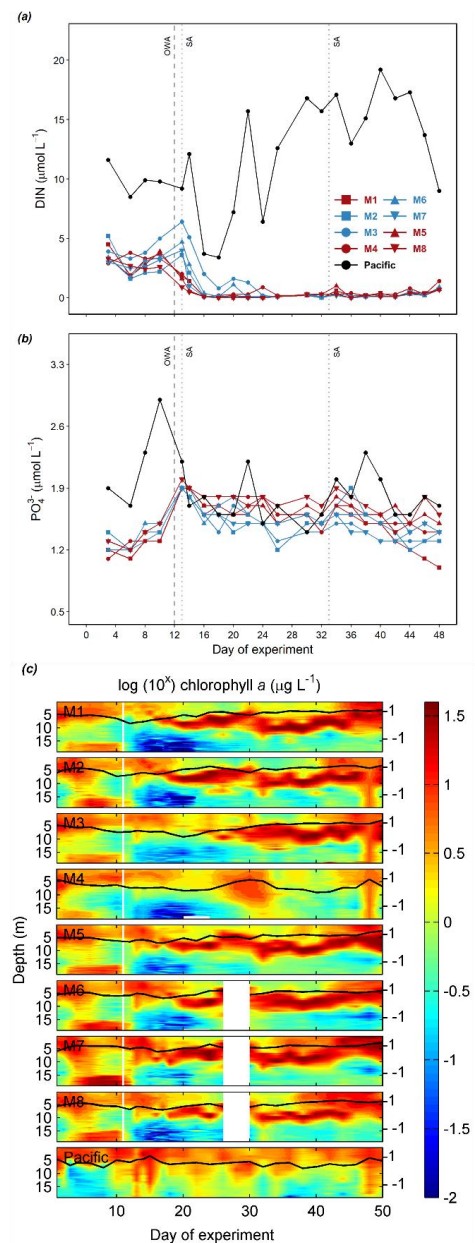


**Figure 2** Temporal dynamics of depth-integrated surface DIN concentration (a), $PO_4^{3-}$ concentration(b) and vertical distribution of chl *a* concentration determined by CTD casts (c). The black solid lines on top of the coloured contours represent the average values over the entire water column, with the corresponding additional y-axes on the right. The vertical white lines represent the day when OMZ water was added to the mesocosms. Color codes and symbols denote the respective mesocosm. Abbreviation: OWA, OMZ water addition. SA, salt addition. Dataset is available at https://doi.pangaea.de/10.1594/PANGAEA.923395 (Bach et al., 2020b).

### 3.2 Temporal dynamics of carbonate chemistry

Before OMZ water addition, surface layer $pH_T$ in the mesocosms ranged between 7.80-7.94 with a slight decline by ~0.1 over

time (Fig. 3a). The initial surface layer TA ranged between 2,310 and 2,330 µmol kg$^{-1}$ (Fig. 3b; day 3-12). Surface layer $pCO_2$

and DIC ranged from 541 to 749 µatm and 2,119 to 2,180 µmol kg$^{-1}$, respectively (Fig. 3c, d).



The two collected OMZ-water masses shared similar carbonate chemistry properties despite the differences in DIN concentrations. In both water masses, $pH_T$ was ~7.48, DIC was ~2,305-2,310 µmol kg$^{-1}$, TA was ~2,337 µmol kg$^{-1}$, and $pCO_2$ was between 1,700 and 1,780 µatm (Table 4).

Table 4: The *in situ* $pH_T$, TA, DIC, $pCO_2$, $\Omega_{Ar}$ and $\Omega_{Ca}$ of the two collected OMZ-water masses.

| Water mass | $pH_T$ | TA (µmol kg$^{-1}$) | DIC (µmol kg$^{-1}$) | $pCO_2$ (µatm) | $\Omega_{Ar}$ | $\Omega_{Ca}$ |
|---|---|---|---|---|---|---|
| Low DIN | 7.49 | 2336.5 | 2305.4 | 1707.5 | 0.90 | 1.38 |
| Very low DIN | 7.47 | 2338.2 | 2312.1 | 1775.3 | 0.87 | 1.34 |

Surface DIC and $pCO_2$ were elevated from ~2,150 µmol kg$^{-1}$ and ~600 µatm to ~2,200 µmol kg$^{-1}$ and ~900 µatm (except M7) by OMZ water addition, respectively, without distinct differences between the two treatments (Mann-Whitney U-Test, p > 0.05; Fig. 3c). Following OMZ water addition, surface $pCO_2$ in the mesocosms decreased quickly and reached minima at 340-500 µatm (except M3 and M4) on day 24 and 26. These minima corresponded with DIC minima at 2,040-2,110 µmol kg$^{-1}$ and $pH_T$ maxima at 7.9-8.1 (except M3 and M4; Fig. 3c, d). After reaching the minima, surface layer $pCO_2$ exhibited a steady increase to 410- 680 µatm from day 24 to day 38 and later declined in M3, M5, and M7 while the rest remained relatively stable until day 42 (Fig. 3c). Interestingly, and unlike the other mesocosms, after OMZ water addition, $pCO_2$ in M3 steadily declined from 928 to 342 µatm until the end of the experiment while that in M4 remained constantly higher than the other mesocosms (> 700 µatm), with a slightly decreasing trend to 645 µatm towards the end of the study (Fig. 3c).

In the Pacific, much lower surface $pH_T$ and higher surface $pCO_2$ and DIC were observed compared to the mesocosms, with an average of 7.7 (7.6-7.8), 1,078 µatm (775 – 1358 µatm) and 2,221 µmol kg$^{-1}$ (2173 – 2269 µmol kg$^{-1}$; minimum to maximum range in parenthesis; Fig. 3c, d), respectively. TA in the Pacific was initially similar to that in the mesocosms, fluctuating between 2,310 and 2,330 µmol kg$^{-1}$, and later decreased to ~2,310 µmol kg$^{-1}$ for the rest of the study.

Surface waters in the mesocosms and the Pacific were always saturated with respect to calcite and aragonite throughout the entire experimental period, with lower values observed in the Pacific (Fig. 4a, c). Bottom waters in the mesocosms and Pacific were always saturated with respect to calcite during the experiment (Fig. 4b) while bottom waters in the Pacific were undersaturated with respect to aragonite before day 13 (0.88-0.99) and had $\Omega_{Ar}$ values slightly above 1.0 for the rest of the study period (Fig. 4d).

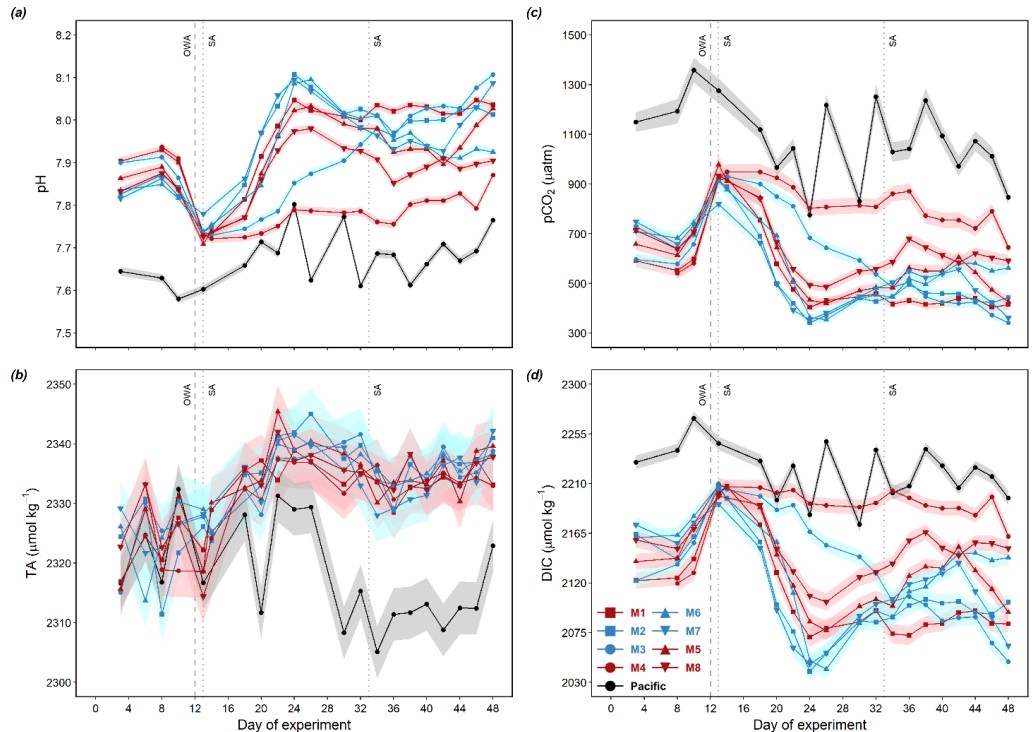

301

**Figure 3 Temporal dynamics of measured depth-integrated surface pH$_T$ (a) and TA (b), and calculated pCO$_2$ (c) and DIC (d). The error ribbons present measurement and propagated standard uncertainties of the calculations, respectively. Color codes and symbols denote the respective mesocosm. Abbreviation: OWA, OMZ water addition. SA, salt addition.**


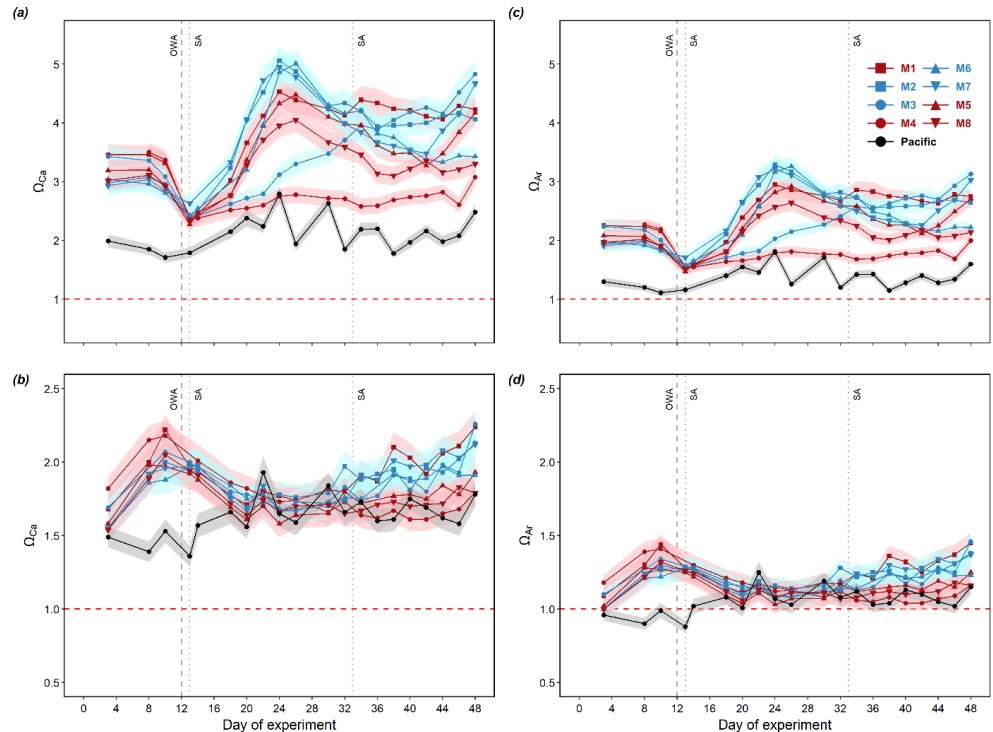

**Figure 4 Temporal dynamics of depth-integrated surface calcite saturation state (a), bottom calcite saturation state (b), surface aragonite saturation state (c), and bottom aragonite saturation state (d) in the mesocosms and the surrounding Pacific. The error ribbons present the propagated standard uncertainties of the calculations. When $\Omega > 1$ (above red dashed line), seawater is supersaturated for calcium carbonate. When $\Omega < 1$ (below red dashed line), seawater is under-saturated for calcium carbonate. Color codes and symbols denote the respective mesocosm. Abbreviation: OWA, OMZ water addition. SA, salt addition.**

### 3.3 Air-sea CO₂ fluxes in the Pacific

Positive $FCO_2$ values indicate $CO_2$ outgassing from the surface waters to the atmosphere, while negative values indicate a $CO_2$ flux from the atmosphere to the ocean. The air-sea $CO_2$ flux in the Pacific was constantly positive throughout our study, fluctuating from 4.2 to 14.0 mmol C m$^{-2}$ d$^{-1}$ over time (Fig. 5a). The minima of $FCO_2$ occurred on day 26 and 30, while the maximum occurred on day 32 when near surface wind was the highest (2.89 m s$^{-1}$; Fig. 5b), corresponding to the minima and maxima of surface $pCO_2$. Co-occurring with a decrease in surface temperature to below 19°C after day 36 (Fig 5c), $FCO_2$ slightly declined from ~10 to ~6 mmol C m$^{-2}$ d$^{-1}$ (Fig. 5a). $FCO_2$ was positively correlated with near surface wind speed ($R^2$ = 0.4). No correlation was found between $FCO_2$ and temperature ($R^2$ = 0).


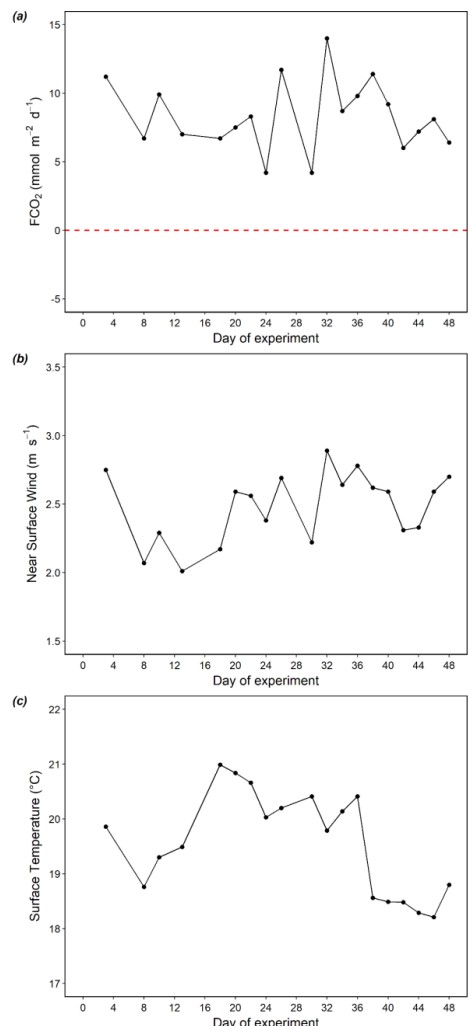

**Figure 5 Temporal dynamics of surface air-sea CO₂ flux (a), near surface wind speed (b) and surface temperature (c) in the Pacific.**
**FCO₂ > 0 (above red dashed line) indicates CO₂ outgassing from the sea surface to the atmosphere. FCO₂ < 0 (below red dashed**
**line) indicates a CO₂ flux from the atmosphere to the sea.**
**4 Discussion**
**4.1 Quality control and propagated uncertainties**
To compare the sensitivity of different calculated variables to uncertainties in the input variables, the propagated uncertainties
were averaged for each calculated variables, reported in numerical values and percentages relative to the calculated values of
each variable (Table 1). Among the 4 reported variables, $\Omega_{Ca}$ and $\Omega_{Ar}$ were the most sensitive to uncertainties in $pH_T$ and TA
with an average uncertainty of 5.1%. This adds ambiguity to whether the bottom water (10-17 m for day 3-28; 12.5-17 m for
day 29-50) in the Pacific was undersaturated with respect to aragonite when $\Omega_{Ar}$ was oscillating near 1 (Fig. 4d). The
propagated uncertainty in $pCO_2$ was slightly lower (3.8%) while DIC was the least sensitive (0.3%).
We examined the internal consistency between DIC measurements and calculations. DIC was measured from day 3 until the
malfunction of the instrument on day 10. In total, 53 sets of measured DIC and calculated DIC (from measured $pH_T$ and TA)
values were obtained from day 3 to day 10 and compared to test their consistency (Fig. 6a). The calculated DIC values were



generally in agreement with the measured values ($R^2$ = 0.985, p < 0.005), showing that the calculations made an overall good
prediction for the measured DIC values. The average of the residuals (calculated DIC– measured DIC) was -8.27 ± 6.9 μmol
kg$^{-1}$, indicating an underestimation of calculated DIC. This result is consistent with a previous observation of underestimated
calculated DIC (pH$_T$, TA) compared with measured DIC when applying the same set of constants (-6.6 ± 7.9 μmol kg$^{-1}$;
Raimondi et al., 2019). The reasons for such underestimation have not been addressed in previous studies and remain unclear.
No significant relationships with input variables pH$_T$ and TA ($R^2$ = 0.12 for both) and temperature ($R^2$ = 0.30) were found in
the DIC residuals (salinity remained the same from day 3 to day 10). The lack of correlation with pH$_T$ and TA indicated that
the underestimation in calculated DIC was not a result from changes in pH$_T$ and TA. Although dissociation constants are
known to be salinity- and temperature-dependent, the lack of correlation between DIC residuals and temperature may be
attributed to the relatively narrow ranges of temperature in the mesocosms (17.9-20.9℃ from day 3-10). The offsets were
typically larger at lower temperatures (e.g., samples from the Arctic, Chen et al. 2015).
To assess the quality of carbonate chemistry measurements in this study, the stability and performance of measurements were
evaluated. The standard deviation of triplicate pH$_T$ measurements varied up to 0.003 with an average of 0.0012 throughout the
whole experiment (Fig. 6b). The average standard deviation was in agreement with reported analytical precisions of pH (0.003,
Orr et al. 2018; 0.002, Raimondi et al., 2019; Ma et al., 2019).
For TA, triplicate measurements of CRM distributed to before and after the sample measurements were carried out on each
measuring day to monitor the stability of the measurement process and the performance of the system. Based on the offsets, a
correction factor was applied to the measured values of samples on each sampling day to calibrate for instrument drift. As
shown in Fig. 6c, 90.5% of the measured TA values of CRM fell between warning limits (UWL and LWL) with one data point
falling outside the control limits (UCL and LCL), overall suggesting a relatively stable measurement system. The average
measured TA was 2209.9 μmol kg$^{-1}$, which was 17.69 μmol kg$^{-1}$ lower than the certified concentration of the CRM (2227.59
μmol kg$^{-1}$), indicating a relatively poor accuracy (compared to the suggested bias of less than 2 μmol kg$^{-1}$; Dickson et al., 2003;
Dickson et al., 2007). The poor accuracy could be attributed to the fact that the concentration of the acid titrant was not checked
after being prepared, as suggested in the protocol (Dickson et al., 2003). A range control chart was computed based on duplicate
measurements of CRM made prior to the sample measurements on each sampling day to evaluate the consistency of the offset
between measured and certified TA values over the course of the study (Fig. 6d; Dickson et al., 2007). The absolute difference
(range) between the repeated CRM mesaurements was on average 1.4 μmol kg$^{-1}$. All the range values fell below the UWL
(3.50 μmol kg$^{-1}$; Fig. 6d), suggesting a relatively good precision of the measurement system.



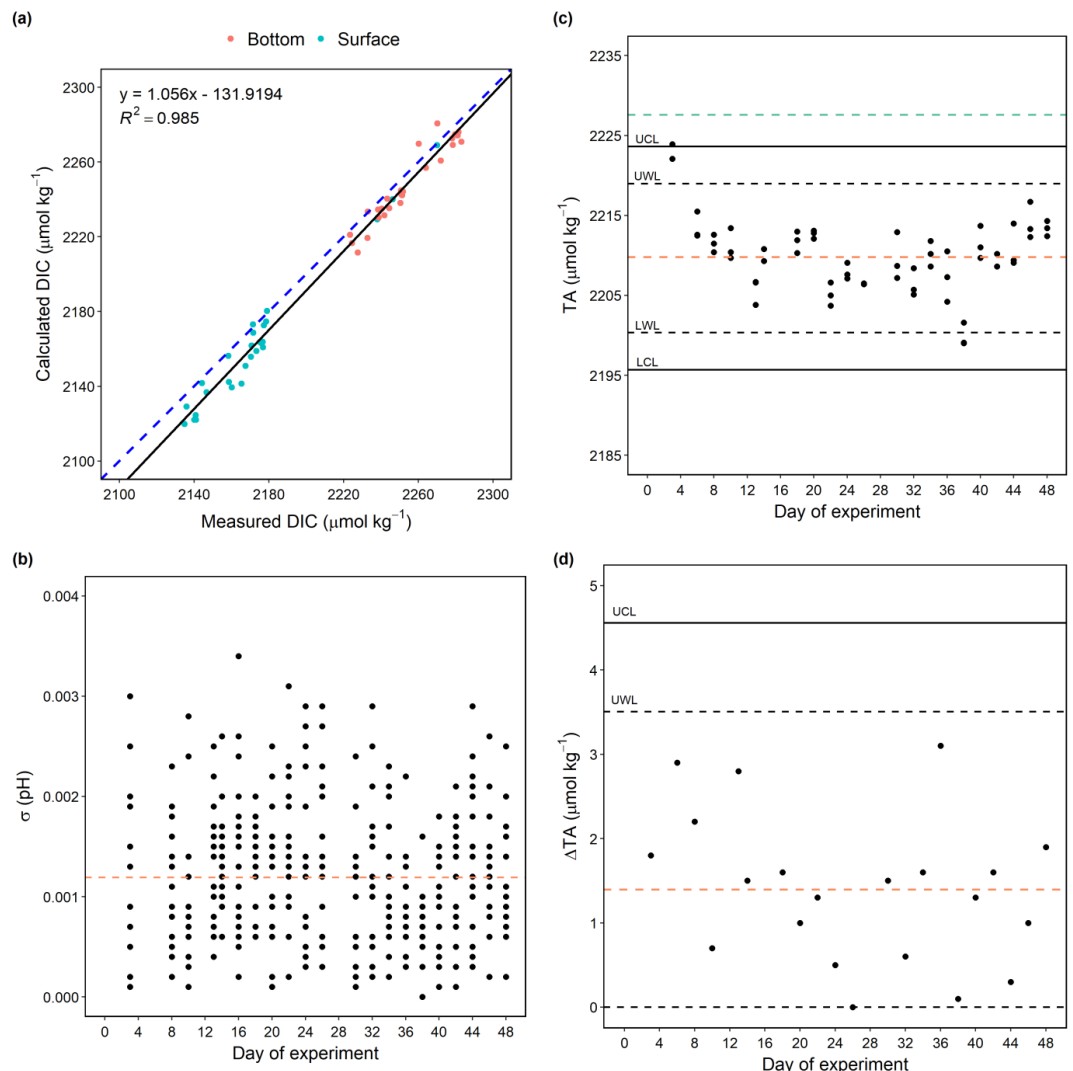

**Figure 6 Comparison of calculated values of DIC (pH$_T$, TA) and measured values (a). The black line is the regression line, with the corresponding equation and R$^2$ shown in the top-left corner. The blue dashed line shows the regression line forced through the origin. Standard deviations of all the triplicate pH$_T$ measurements on each sampling day over the study period. Orange dashed line shows the average (n = 377) of the standard deviations (b). TA values of CRM measurements on each sampling day over the study period. Orange dashed line shows the average (n = 62) of the measured values and green dashed line indicates the certified value of the CRM (c). The absolute difference in TA values between duplicate CRM measurements (range) on each sampling day over the study period. Orange dashed line shows the average (n = 21) of the ranges (d). Abbreviation: UCL, upper control limit. UWL, upper warning limit. LWL, lower warning limit. LCL, lower control limit.**

## 4.2 CO$_2$ responses to the simulated upwelling event

At the beginning of the experiment, surface pCO$_2$ levels in the mesocosms were >500 μatm (Fig. 3c). This suggests that we initially enclosed an upwelled water mass that was enriched with respiratory CO$_2$. The addition of OMZ water with high concentrations of CO$_2$ to the mesocosms reduced the surface pH$_T$ by 0.1-0.2 and increased the surface pCO$_2$ to >900 μatm (except for M7, which was 819.4 μatm on day 13). The simulated upwelling substantially reduced the variability in CO$_2$ between mesocosms because OMZ water addition replaced ~20 m$^3$ of seawater in each mesocosm (out of ~54 m$^3$). The enhanced pCO$_2$ level is comparable with our observations in the ambient Pacific water (>775 μatm; Fig. 3c). These values also



agree with reported observations for our study area in 2013 (>1,200 µatm in the upper 100 m and > 800 µatm at the surface;
Bates, 2018).
In the days after OMZ water addition, surface $pCO_2$ in the mesocosms dropped near or below the atmospheric level (405.22
µatm, NOAA/GML) with a decline in DIC by ~100 µmol kg$^{-1}$ (except M3 and M4; Fig. 3c, d). The declining $pCO_2$ could be
partially attributed by $CO_2$ outgassing due to a high $CO_2$ gradient from the sea surface to the air. Due to a rare coastal El Nino
event (Garreaud, 2018), the $CO_2$ loss process may have been enhanced by a rapid surface warming (19.8-21.0 °C from day 14
to 36; Fig. 5) which reduced surface $CO_2$ solubility (Zeebe and Wolf-Gladrow, 2001). However, air-sea gas exchange could
not explain surface $CO_2$ under-saturation in relation to the atmosphere, as observed in response to OMZ water addition in some
mesocosms (Van Geen et al., 2000; Friederich et al., 2008; Fig. 3c). Biological production has typically one to four times
greater impacts on $CO_2$ drawdown than air-sea gas exchange in the equatorial Pacific where surface waters are exposed to
local wind stress (Feely et al., 2002). This interpretation is supported by the continuously high DIC in M4 where photosynthetic
biomass build-up was substantially lower (Fig. 3d). Hence, the depletion of nutrients (Fig. 2a, b) and increase in chl *a*
concentration (Fig 2c; Bach et al., 2020a) strongly suggest that the loss of DIC (except M4) was primarily driven by biological
uptake and phytoplankton growth. Nevertheless, it is difficult to dissect how much $CO_2$ was outgassed and how much was
taken up photosynthetically as we did not measure air-sea gas exchange in the mesocosms (please note that equations from
Wanninkhof, (2014) are not applicable for mesocosms).
Before OMZ water addition, dissolved inorganic N:P ratios in the mesocosms ranged from 1.6 to 3.5 (data not shown),
indicating N is the limiting nutrient in the water column (Bach et al., 2020a). Not surprisingly, the uptake of DIC was higher
in the "Low DIN" mesocosms which received more input of DIN from OMZ water addition, with on average 41.0 µmol kg$^{-1}$
higher drawdown compared to the "Very Low DIN" from day 13 to day 24 (excluding M3 and M4; Mann-Whitney U-Test, p
= 0.05; Table S1). This observation agrees with the general expectations that addition of limiting nutrients to water column
should enhance biological biomass build-up. Such differences in DIC uptake, however, were not reflected in the build-up of
particulate organic carbon (POC) in the mesocosms (excluding M3 and M4; Mann-Whitney U-Test, $p > 0.1$). As mentioned
above, the differences in OMZ-water DIN between the two treatments were minor and hence, their potential to trigger treatment
difference were small. Due to the developing N-limitation after the biomass build-up there much of the consumed DIC could
have been channelled to dissolved organic carbon (DOC) pool. Indeed, we observed a pronounced increase in DOC following
OMZ water addition (except for M4; Igarza et al., in prep, 2021). The increase in DOC may be attributed to extracellular
release by phytoplankton due to nutrient limitation, or cellular lysis of phytoplankton cells by bacteria (Myklestad 2000; Igarza
et al., in prep, 2021).
After day 24, variability in carbonate chemistry between individual mesocosms increased, with a general trend of recovering
from $CO_2$-undersaturated conditions during the peak of the bloom (except for M3 and M4; Fig. 3c). One factor that may have
controlled the differences in $CO_2$ increase are the mesocosm-specific phytoplankton succession patterns. A shift from a diatom-
dominated community to a dominance of dinoflagellates (in particular *Akashiwo sanguinea*) occurred when DIN was
exhausted, which was absent in M3 and M4 (Bach et al., 2020a). The different succession patterns in the plankton community
are the most likely explanation why M3 and M4 behaved differently from the others in terms of surface layer productivity, and
hence carbonate chemistry. Although the rate of DIN depletion in M3 and M4 were similar to the others, the reduction in $pCO_2$
in M3 experienced a 1-week delay which is consistent with the delayed build-up of chl *a* biomass (Fig. 2c, 3c). On the other
hand, the $pCO_2$ level in M4 remained constantly elevated throughout the experiment, as of a lack of a phytoplankton bloom
(Fig. 2c, 3c). M4 was the only mesocosm where a *A. sanguinea* remained undetectable, whereas a delayed and reduced
contribution by *A. sanguinea* was observed in M3. This strongly suggests that *A. sanguinea* was a key factor driving the trend
of carbonate chemistry in the mesocosms.





Near the end of the experiment, a slight decline in pCO₂ became apparent in the mesocosms which co-occurred with a second
phytoplankton bloom observed in the uppermost layer of the water column (Fig. 2c, 3c). This bloom was likely fuelled by
surface eutrophication due to defecating sea birds. During the last part of our experiment, Inca terns (*Larosterna inca*) were
frequently observed to rest on the roofs and the edges of the mesocosms (Bach et al., 2020a). Bird excrements, dropped into
the mesocosms, are known to be enriched in inorganic nutrients, especially ammonium (Bedard et al., 1980). The excrements
may also be high in dissolved organic nitrogen (DON), evidenced by a substantial increase in DON concentrations in the
mesocosm surface from day 38 onward (Igarza et al., in prep, 2021). The triggered surface eutrophication and phytoplankton
blooms were noticeable from an accumulation of chl *a* biomass above the mixed layer in the mesocosms near the end of the
study (Fig. 2c). As a result, another drawdown of DIC could be observed in the mesocosms except for M4, M6 and M8. While
the build-up of chl *a* was comparable with that triggered by OMZ water addition, the drawdown in DIC was less pronounced,
potentially counteracted by the release of CO₂ by enhanced respiration and remineralization following the previous bloom.
Also, the second bloom occurred in the top 2 meter in the mesocosms (Fig. 2c) where gas exchange can quickly replete the
DIC drawdown during photosynthesis and biomass build up.

### 4.3 Temporal changes of carbonate chemistry in the coastal Pacific near Callao

According to estimations by Takahashi et al. (2009) of global air-sea CO₂ fluxes, our study site in the equatorial Pacific (14°N-
14°S) is a major source of CO₂ to the atmosphere. Our near-coastal location showed high pCO₂ levels over the study period
(with an average of 1,078 µatm), with a sea-to-air CO₂ flux of 4.2-14.0 mmol C m⁻² d⁻¹ (Fig. 5). Compared to the criterion of
high CO₂ fluxes (5 mmol C m⁻² d⁻¹ or more) as proposed by Paulmier et al. (2008), our study site was a strong CO₂ source to
the atmosphere most of the time. These results of air-sea CO₂ fluxes were slightly higher than observations by Friederich et al.
(2008) along the coast of Peru in February, 2004-2006 (0.85-4.54 mol C m⁻² yr⁻¹; spatially averaged for 5-15°S along the coast
of Peru). This is not surprising because Friederich et al. averaged the air-sea CO₂ fluxes for 0-200 km from shore where much
lower pCO₂ were observed offshore (< 600 µatm), compared to our nearshore study site. The decline in pCO₂ with increasing
distance from shore was driven by biological uptake and outgassing to the atmosphere (Friederich et al., 2008; Loucaides et
al., 2012). However, when compared to the magnitude of DIC drawdown triggered by upwelling events in the mesocosms, the
flux of CO₂ to the atmosphere was insignificant. Assuming a 10 m mixed layer in the Pacific with a DIC concentration of
2,200 µmol kg⁻¹, the DIC content below 1 m² surface area would be ~22 mol m⁻². With an upper bound outgassing of 14.2
mmol C m⁻² d⁻¹ over 10 days (day 13-24), the loss of CO₂ would only be 0.142 mol m⁻². On the other hand, the average DIC
drawdown of 118.2 µmol kg⁻¹ in the "Very Low DIN" and 160.3 µmol kg⁻¹ in the "Low DIN" mesocosms (M3 and M4
excluded) during this period accounts for 1.18 mol m⁻² and 1.60 mol m⁻², respectively, over the same water column. This shows
that biological processes, drawing down CO₂, is stronger than loss by air-sea gas exchange.
During our study, we experienced a coastal El Niño event, which has been the strongest on record (compared to those recorded
in 1891 and 1925) and induced rapid sea surface warming of ~1.5℃ and enhanced stratification (Garreaud, 2018). Previous
investigations showed that the impact of reduced upwelling on CO₂ fluxes is pronounced for upwelling areas (Feely et al.,
1999; Feely et al., 2002). A decline in upwelling of CO₂-enriched OMZ water results in a decrease in sea-to-air CO₂ fluxes.
For example, during the 1991-94 El Niño year, a total reduction in CO₂ fluxes to the atmosphere was reported for the equatorial
Pacific. They were only 30-80% of that of a non-El-Niño year (Feely et al., 1999; Feely et al., 2002). This is likely to be the
case for our study location. Most studies investigated air-sea CO₂ fluxes at larger time and regional scales (Feely et al., 1999;
Friederich et al., 2008; Takahashi et al., 2009). Therefore, it is difficult to conclude the magnitude of the coastal El Niño
influence on the local CO₂ fluxes in our study by comparing our results with previous observations. Nevertheless, our
observations can serve as a first evidence of carbonate chemistry dynamics in the coastal Peruvian upwelling system during a
coastal El Niño event. Observations of sea surface carbonate chemistry with a high temporal resolution (every-2ⁿᵈ-day) in near-



shore waters are scarce, as rarely covered by typical research expeditions in the open ocean (Takahashi et al., 2009; Franco et
al., 2014), especially during such an extremely rare coastal El Niño event. Comparisons of our data with previous or future
observations may enhance our understanding of how inorganic carbon cycling interact with extreme climate events in
upwelling systems.
$CO_2$-enriched OMZ water has been occasionally reported to be under-saturated with respect to aragonite (Feely et al., 2008;
Fassbender et al., 2011). In our study, calcite under-saturation did not occur in the mesocosms or in the Pacific (Fig. 4).
Aragonite under-saturation, however, was observed below the surface (10-17 m for day 3-28; 12.5-17 m for day 29-50) of the
Pacific at the start of the experiment (Fig. 4d), when $pCO_2$ was the highest ($pCO_2 > 1100$ µatm; Fig. 3c). Aragonite under-
saturation was also observed in the two deep water masses collected at deeper depths (30 m and 70 m) in the Pacific (Table
4). Throughout the study period, the aragonite saturation state fluctuated close to around 1 below the surface (Fig. 4d).
Considering the water column we sampled in the Pacific still belonged to the upper surface ocean, we could expect deeper and
more $CO_2$-enriched water in the underlying OMZ to be most likely under-saturated with respect to calcite and aragonite. Hence,
our observations of aragonite under-saturation in the Pacific suggest a potential risk of dissolution for marine calcifiers in
response to the on-going intensification and expansion of acidified OMZ water (Comeau et al., 2009; Lischka et al., 2011;
Maas et al., 2012).
**5 Conclusion**
Our observations in the mesocosms revealed that, following the addition of two OMZ water masses with different nutrient
signatures, there was a higher drawdown of DIC in response to slightly more DIN input from the OMZ water addition but no
difference in the build-up of POC and chl *a* (Fig. 2a, 2c, 3d). The timing of the first phytoplankton bloom was consistent with
a shift from a diatom-dominated community to *A. sanguinea* dominance in most mesocosms, indicating that *A. sanguinea* was
a key factor driving the changes in carbonate chemistry under N-limited conditions. A second phytoplankton bloom was
triggered by defecations of Inca terns, which eased the N limitation in the mesocosms (Fig. 2c). These findings provide
improved insights into the links between upwelling-induced N limitation, phytoplankton community shifts and carbonate
chemistry dynamics in the Peruvian upwelling system.
The surrounding Pacific waters at the study site were characterized by constantly high $pCO_2$ levels (with an average of 1,078.1
µatm). Most $CO_2$ flux estimates have been conducted in the open ocean and few studies surveyed coastal regions (Takahashi
et al., 2009; Franco et al., 2014). Our study site was a strong $CO_2$ source to the atmosphere most of the time (4.2-14.2 mmol
C m$^{-2}$ d$^{-1}$), despite a rare coastal El Niño event. However, evidence from our mesocosm experiment suggests biological
responses that draw down DIC can quickly turn a $CO_2$ source into a sink in the upwelling system. The influence of the co-
occurring coastal El Niño event on the local $CO_2$ fluxes remains unclear. Nevertheless, future carbonate chemistry fluctuations
are expected to be enhanced by expanding and intensifying ocean deoxygenation, as well as reducing buffer factors (Schulz et
al., 2019). Hence, it is essential to improve our understanding of the mechanisms driving the inorganic carbon cycling in
upwelling systems. As a unique dataset that characterized near-shore carbonate chemistry with a high temporal resolution
during a rare coastal El Niño event, our study gives important insights into the carbonate chemistry responses to extreme
climate events in the Peruvian upwelling system.
**Data availability**
All data will be made available on the permanent repository www.pangaea.de after publication.



**Author contribution**

UR, KGS, and LTB designed the experiment. All authors contributed to the sampling. S-MC measured, calculated, and analyzed carbonate chemistry. LTB and KGS supervised the carbonate chemistry analysis. KGS carried out the CTD casts and data analyses. EvdE and EPA measured and analyzed nutrients. S-MC wrote the manuscript with input from all the co-authors.

**Competing interests**

The authors declare that they have no conflict of interests.

**Acknowledgements**

This project was supported by the Collaborative Research Centre SFB 754 Climate-Biogeochemistry Interactions in the Tropical Ocean financed by the German Research Foundation (DFG). Additional funding was provided by the EU project AQUACOSM and the Leibniz Award 2012 granted to U.R. We thank all participants of KOSMOS Peru 2017 experiment for mesocosm maintenance and sample collection and analysis. Special thanks go to the staff of IMARPE, the captains and crews of *Bap Morales*, *IMARPE VI* and *B.I.C. Humboldt*, and Marina de Guerra del Perú, in particular the submarine section of the Navy of Callao, and the Dirección General de Capitanías y Guardacostas for their support and assistance planning and carrying out the experiment. We are thankful to Club Náutico Del Centro Naval for hosting our laboratories, office space, and support. This work is a contribution in the framework of the Cooperation agreement between the IMARPE and GEOMAR through the German Ministry for Education and Research (BMBF) project ASLAEL 12-016 and the national project Integrated Study of the Upwelling System off Peru developed by the Direction of Oceanography and Climate Change of IMARPE, PPR 137 CONCYTEC. Analyses and visualizations used in this paper were produced with the Giovanni online data system, developed and maintained by the NASA GES DISC.

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
