# Peer review of "Temporal dynamics of surface ocean carbonate chemistry in response"

_Biogeosciences, 2021_

## Author Response (AR1)

**Response to Reviewers**

We thank both reviewers for their insightful comments and suggestions. Please find below our detailed point-to-point responses.

**Reviewer #1**

"The study describes the response of the carbonate system to a simulated upwelling event in large mesocosms installed off the Peruvian coast.

This is a well written and carefully executed study using state of the art methods. The authors did an excellent job describing the experimental design and addressing the uncertainties of their key parameters, pH and TA. This is a pretty straight forward manuscript in a well-designed experimental framework producing important quantitative results. The observed changes in air-sea CO2 fluxes and the carbonate saturation state supply the scientific community with important information. These results help understanding the role of productive coastal upwelling systems for CO2 exchange and ocean acidification in response to climate change and associated extensions of oxygen minimum zones and more frequent extreme weather events.

The authors carefully addressed measurement problems and uncertainties of their estimates and I see no major issue with their experimental approach, analytical methods or data interpretation."

RESPONSE: We thank the Reviewer for reviewing our manuscript and the kind words.

**Reviewer #2**

**General comments**

1. "This is a well written article describing an interesting and relevant study. I found the article to be a little imbalanced with respect to a discussion of the uncertainties and potential complications in the interpretation of the results. For example, there was a very thorough discussion of the uncertainties in the chemical measurements and carbon system calculations, but almost no discussion of the potential complications introduced by the brine addition or the mesocosms themselves. I know that these mesocosms have been used for many years and have been well described, but it still would have been useful to at least mention some of the potential issues associated with some of the major findings (e.g. how might the setup itself have contributed to the change in ecosystem structure)."

RESPONSE: We thank the reviewer for the kind words. This paper specifically focuses on carbonate chemistry dynamics in the Peruvian upwelling system and hence it is important to address the uncertainties in the $CO_2$ system measurements and calculations. The changes in the ecosystem structure and the impacts of mesocosm setup (heterogenous initial conditions in the mesocosms, light limitation via self-shading, etc.) were extensively discussed in the overview paper by Bach et al. (2020), so we think it is not necessary to repeat this discussion in this closely related paper.

With respect to the brine addition: the reviewer is right that the brine addition (and other mesocosm specific manipulations including the enclosure itself) could have influenced the community composition. This is difficult to investigate because the manipulations were done in all the mesocosms at the same time so we cannot determine how it would have been without any manipulations. Nevertheless, the effects of the brine on the enclosed organisms have been discussed in past studies and considered negligible for a salinity increase of less than 1 (~0.7 and ~0.5 increase for both salt additions respectively; Czerny et al., 2013a). The difference of salinity in the mesocosms from the Pacific was less than 1 throughout the study so we believe  salinity was not a stressor to the system (Bach et al., 2020). However, we acknowledge that an effect cannot fully be excluded for the Peru study and have disclosed that in the revised version of the manuscript. Other limitations are thoroughly discussed by Bach et al. (2020) and we have refered to this paper when necessary. One limitation of the mesocosm setup that is specific to the carbonate chemistry development is how the enclosure affects air-sea gas exchange. We have expanded the discussion on this aspect in the revised manuscript.

2. "I also think the authors could have tried other approaches to try and separate the gas exchange carbon loss from the biological uptake. Related to this, I was surprised that there was no discussion of the dissolved oxygen in the mesocosms. The methods indicate that DO was part of the CTD package. Since the article is about OMZ waters, I would have expected a section on oxygen changes. DO can also help clarify the biological versus gas exchange losses."

RESPONSE: A common practice to monitor gas exchange in the mesocosms is $N_2O$ addition (Czerny et al., 2013b). However, this was not carried out in our study because it may interfere with [15]N label incubations (Schulz et al., 2021). This has been clarified in the revised manuscript. We agree that a discussion of DO changes would be necessary for such a large-scale experiment carried out in OMZ waters. However, this would be outside the scope of this paper and has been discussed in other papers on other aspects from the same experiment (Bach et al., 2020; Schulz et al., 2021). The impacts of DO are more pronounced for N-cycling processes and therefore were specifically addressed by Schulz et al. (2021). Although DO and $CO_2$ are usually negatively correlated in OMZs, they are mainly driven by microbial respiration below the surface. Without knowledge of gas exchange which was not

measured and could not be calculated in our study, it is impossible to clarify the relative importance of biology versus gas exchange losses from DO to $CO_2$ changes. In this paper, we focus on the surface ocean carbonate chemistry which is mainly driven by the N deficit and enriched $CO_2$ of the upwelled OMZ waters, and hence, we will not repeat the discussion of DO here.

**Specific comments**

3. **Line 39** – "technically the denitrification and anammox processes do not remove nitrogen from the ocean…at least without considering the required mixing to the surface and gas exchange. It would be more correct to say that these processes remove biologically available nitrogen."

RESPONSE: Thanks for pointing this out. We have changed "nitrogen" to "biologically available nitrogen".

4. **Line 45** – "There is no "uptake" of anthropogenic atmospheric CO2 in these high CO2 upwelling waters. You may have less degassing of the waters because of elevated atmospheric CO2, but that is not the same as uptake of anthropogenic atmospheric CO2."

RESPONSE: The revised sentence in the manuscript is "Apart from being N-deficient, the OMZ waters are also characterized by enhanced carbon dioxide ($CO_2$) concentrations and low pH from respiratory processes and are further acidified by increasing uptake of anthropogenic atmospheric $CO_2$ (Feely et al., 2008; Friederich et al., 2008; Paulmier et al., 2008; Paulmier et al., 2011)."

5. **Line 105:129** – "I understand the concept behind the brine additions, but it seems that this could affect the ecosystem structure in the mesocosms. How do the authors know that this artificial halocline did not impact the results? In a similar vein, upwelling not only brings up high CO2 and high nutrient waters, but it also brings up colder waters. I assume the water added to the mesocosms was not temperature controlled, so how might that affect the results?"

RESPONSE: Please see our reply to general comment (1) for brine additions. It is true that the upwelling brings up cool water. But because the mesocosms were deployed "in situ" they had the average temperature of the surrounding Pacific, which carried the average temperature of the cool upwelled water and the warmer surface water. Therefore, the mesocosms were in a way temperature controlled as they represented the real-world temperature profile in the upwelling areas. The temperature developments inside the mesocosms and in the Pacific surrounding them are shown in the overview paper by Bach et al. (2020).

6. **Line 232** – "unnecessary underlining"

RESPONSE: The underline has been removed.

7. **Line 391:393** – "The authors say that it is difficult to determine how much CO2 was lost due to gas exchange because the Wanninkhof flux equations do not work in mesocosms. Since the mesocosms are essentially closed systems (except for exchange with the atmosphere), why can't the authors do a water carbon budget at the beginning of the experiment and the end to determine how much carbon was lost from the system? With the change in water chemistry and quantification of the particulates in the sediment trap, the change in total carbon should reflect the loss to the atmosphere."

RESPONSE: Please see our reply to general comment (2). Carbon budget is an approach that we always attempt to do for mesocosm experiments. However, due to high variability of DOC data and the poorly constrained gas exchange of $CO_2$, this approach often comes with high uncertainties and large errors even if we have a relatively simple dataset (Czerny et al., 2013b; Boxhammer et al., 2018). It becomes even more difficult for the current dataset because the water column was not homogenously mixed like in previous studies. With a lack of estimation of $CO_2$ gas exchange, this approach was unsuccessful to calculate a reasonable C budget in this study. We have addressed this issue in the revised manuscript.

8. **Line 403:406** – "How might the lack of POC buildup be related to the artificial halocline created from the brine addition? Would this keep the particles from settling out, allowing them to be recycled more effectively than one would observe in the natural environment?"

RESPONSE: I think there may be misunderstanding in terms of POC buildup. There was POC buildup in most mesocosms in response to the OMZ water addition (except M4), which did not reflect the difference in DIC uptake between treatments. Since a brine addition was performed to all the mesocosms and M4 was the only mesocosm lacking POC buildup, the brine addition is unlikely the reason for the absence of POC buildup in M4. Instead, a *A. sanguinea* (a mixotrophic dinoflagellate) bloom did not occur in M4 which made it different from the other mesocosms (Bach et al., 2020). *A. sanguinea* was persistent in the water column in the mesocosms (except M4 where it never bloomed) and they retained the biomass in the water column, so they did not sink out until the end of the experiment (Bach et al., 2020). The lack of *A. sanguinea* bloom in M4 may be attributed to the variable initial conditions in the mesocosms that led to divergent plankton succession patterns in the mesocosms. We have clarified this in the revised manuscript.

9. **Line 407:408** – "By recovering from CO2-undersaturation, do the authors mean that the waters were taking up CO2 from the atmosphere? I am surprised that there has been no discussion of oxygen concentrations up to this point. Would dissolved oxygen help sort out the biological from gas exchange components?"

RESPONSE: By $CO_2$-undersaturation, we refer to the finding that intense primary production can deplete surface $CO_2$ below atmospheric equilibrium and stop $CO_2$

outgassing (Van Geen et al., 2000; Friederich et al., 2008; Loucaides et al., 2012). For the reasons why we did not further consider DO developments, please see our reply to general comment (2).

10. **Line 420:421** – "The bird droppings are unfortunate. The authors raise the issue of nutrient addition, but I wonder if they could potentially impact the pH of the system?"

RESPONSE: Good point. Seabird excrement generally contains 60% water, 7.3% N and 1.5% P and the main form of N is uric acid and ammonium which makes it slightly acidic (De La Peña-Lastra, 2021). Therefore, the droppings may lower the pH of the surface water. However, this was not visible from our observations and was likely counteracted by the pH increase due to the guano-triggered primary production. This information has been added in the revised manuscript.

11. **Line 478:480** – How do the authors know that the change in ecosystem structure resulted from the change in nutrients with the upwelled water and not from the change in hydrodynamics (e.g. mixing) within the mesocosm? Is there evidence in the coastal ocean of similar changes in structure associated with the upwelling?

RESPONSE: The reviewer raised a valid point. It would be ideal to have control mesocosms that were treated the same way except the OMZ water addition (but surface water addition instead) to rule out the effects induced by hydrodynamics. This has been compromised to ensure enough replicate numbers for both treatments despite the enormous cost of mesocosm experimentation. Nevertheless, a previous study has examined impacts of different mixing techniques in outdoor mesocosms and found no effects on phytoplankton biomass and minor effects on phytoplankton and zooplankton community composition (Striebel et al., 2013). In our study, various measures were also taken to minimize the mixing (brine additions, slow casting of CTD, etc.). We have addressed this issue in the revised manuscript. As far as we know, this is the first large-scale mesocosm study in the Peruvian upwelling system and therefore evidence from a similar experimental setup is scarce, if not absent. More specialized papers will be published within the special issues to provide more details on other biogeochemical and ecological aspects in the Peruvian upwelling system during the coastal El Niño.

**References**

Bach, L. T., Paul, A. J., Boxhammer, T., von der Esch, E., Graco, M., Schulz, K. G., Achterberg, E., Aguayo, P., Arístegui, J., Ayón, P., Baños, I., Bernales, A., Boegeholz, A. S., Chavez, F., Chavez, G., Chen, S.-M., Doering, K., Filella, A., Fischer, M., Grasse, P., Haunost, M., Hennke, J., Hernández-Hernández, N., Hopwood, M., Igarza, M., Kalter, V., Kittu, L., Kohnert, P., Ledesma, J., Lieberum, C., Lischka, S., Löscher, C., Ludwig, A.,

Mendoza, U., Meyer, J., Meyer, J., Minutolo, F., Ortiz Cortes, J., Piiparinen, J., Sforna, C., Spilling, K., Sanchez, S., Spisla, C., Sswat, M., Zavala Moreira, M., and Riebesell, U.: Factors controlling plankton community production, export flux, and particulate matter stoichiometry in the coastal upwelling system off Peru, Biogeosciences, 17, 4831–4852, https://doi.org/10.5194/bg-17-4831-2020, 2020.

Boxhammer, T., Taucher, J., Bach, L.T., Achterberg, E.P., Algueró-Muñiz, M., Bellworthy, J., Czerny, J., Esposito, M., Haunost, M., Hellemann, D. and Ludwig, A.: Enhanced transfer of organic matter to higher trophic levels caused by ocean acidification and its implications for export production: A mass balance approach. PloS one, 13(5), p.e0197502, 2018.

Czerny, J., Schulz, K.G., Krug, S.A., Ludwig, A. and Riebesell, U.: The determination of enclosed water volume in large flexible-wall mesocosms "KOSMOS", Biogeosciences, 10(3), 1937-1941, https://doi.org/10.5194/bg-10-1937-2013, 2013a.

Czerny, J., Schulz, K. G., Ludwig, A., and Riebesell, U.: Technical Note: A simple method for air–sea gas exchange measurements in mesocosms and its application in carbon budgeting, Biogeosciences, 10, 1379–1390, https://doi.org/10.5194/bg-10-1379-2013, 2013b.

De La Peña-Lastra, S.: Seabird droppings: Effects on a global and local level, Science of The Total Environment, Feb 1;754:142148, https://doi.org/10.1016/j.scitotenv.2020.142148, 2021

Friederich, G.E., Ledesma, J., Ulloa, O. and Chavez, F.P.: Air–sea carbon dioxide fluxes in the coastal southeastern tropical Pacific, Prog. Oceanogr., 79(2-4), 156-166, https://doi.org/10.1016/j.pocean.2008.10.001, 2008.

Igarza, M.,  Sánchez, S., Bernales, A., Gutiérrez, D., Meyer, J., Riebesell, U., Graco, M., Bach, L., Dittmar, T., and Niggemann, J.: Dissolved organic matter production during an artificially-induced red tide off central Peru. Biogeosciences, in preparation, 2021.

Loucaides, S., Tyrrell, T., Achterberg, E.P., Torres, R., Nightingale, P.D., Kitidis, V., Serret, P., Woodward, M. and Robinson, C.: Biological and physical forcing of carbonate chemistry in an upwelling filament off northwest Africa: Results from a Lagrangian study, Global Biogeochem. Cycles, 26(3), https://doi.org/10.1029/2011GB004216, 2012.

Myklestad, S.M.: Dissolved organic carbon from phytoplankton, In Mar. Chem., 111-148, Springer, Berlin, Heidelberg, 2000.

Schulz, K. G., Achterberg, E. P., Arístegui, J., Bach, L. T., Baños, I., Boxhammer, T., Erler, D., Igarza, M., Kalter, V., Ludwig, A., Löscher, C., Meyer, J., Meyer, J., Minutolo, F., von der Esch, E., Ward, B. B., and Riebesell, U.: Nitrogen loss processes in response to upwelling in a Peruvian coastal setting dominated by denitrification – a mesocosm

approach, Biogeosciences, 18, 4305–4320, https://doi.org/10.5194/bg-18-4305-2021, 2021.

Striebel, M., Kirchmaier, L. and Hingsamer, P.: Different mixing techniques in experimental mesocosms—does mixing affect plankton biomass and community composition?, Limnology and Oceanography: Methods, 11(4), 176-186, https://doi.org/10.4319/lom.2013.11.176, 2013.

Van Geen, A., Takesue, R.K., Goddard, J., Takahashi, T., Barth, J.A. and Smith, R.L.: Carbon and nutrient dynamics during coastal upwelling off Cape Blanco, Oregon, Deep Sea Res. Part II Top. Stud. Oceanogr., 47(5-6), 975-1002, https://doi.org/10.1016/S0967-0645(99)00133-2, 2000.